# The Effect of Tillage on Faba Bean (*Vicia faba* L.) Nitrogen Fixation in Durum Wheat ((*Triticum turgidum* L. subsp. *Durum* (Desf))-Based Rotation under a Mediterranean Climate

Luigi Tedone *, Salem Alhajj Ali [ID] and Giuseppe De Mastro [ID]

Department of Soil, Plant and Food Sciences, University of Bari "Aldo Moro", Via G. Amendola 165/a, 70126 Bari, Italy
* Correspondence: luigi.tedone@uniba.it; Tel.: +39-0805442501

**Abstract:** Biological nitrogen fixation (BNF) is a sustainable approach to improving soil fertility that not only provides nitrogen to subsequent crops but also reduces the impacts of synthetic fertilizers. Here, a field experiment was established within the faba bean (*Vicia faba* L.), cv Prothabat 69-durum wheat (*Triticum turgidum* L. subsp. *Durum* (Desf)), cv Iride rotation framework of a long-term experiment in southern Italy to quantify BNF over two consecutive years (2012/13–2013/14). The effect of tillage systems (reduced, conventional, and no tillage) on faba bean $N_2$ fixation was estimated at the flowering and maturity stages via the natural abundance technique, using wheat as a reference crop. The effect of tillage on the percentage of nitrogen fixation from the atmosphere (Ndfa) and the amount of $N_2$ fixed (kg ha$^{-1}$) were higher under a no-tillage system in both years and at both growth stages, with values of 66.5% at flowering and 81.7% at maturity. The same trend was reported for the amount of $N_2$ fixed (kg N ha$^{-1}$) at both faba bean growth stages. The N balance was positive in both years, with a mean value of 40.4 kg N ha$^{-1}$, across all tillage systems; this value was greater in the no-tillage systems (45.7 kg N ha$^{-1}$) with respect to the others. The values for the organic matter content and stability index were higher under the no-tillage system, which provided favourable conditions that improved $N_2$ fixation by faba beans. The overall results indicate that no-tillage soil management represents a sustainable strategy for improving soil quality and fertility, therefore reducing the dependency of agriculture on synthetic fertilizers.

**Keywords:** nitrogen fixation; faba bean; no tillage; soil organic matter; crop rotation

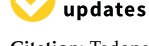



## 1. Introduction

Biological nitrogen fixation (BNF) is an important natural process involving particular legume species where atmospheric nitrogen ($N_2$) is fixed through the symbiosis between plants and bacteria and converted into ammonia (NH3) in plant tissues. BNF represents an environmentally friendly source of N for agricultural soils, as a fundamental component for plant growth and production, that can be industrially produced in several ways [1,2]. BNF is of great practical importance due to its positive effects on several soil properties; for example, it can promote an increase in nutrient availability for crops, reducing the use of synthetic fertilizers that contribute to severe environmental impacts. It is widely accepted that annual legume crops, grown in rotation with cereal crops, can contribute to N and phosphorus availability in the soil [3], providing a significant amount of biologically fixed nitrogen for the subsequent crops and therefore improving crop productivity and yield [4]. Kopke and Nemecek [2] reviewed the literature on symbiotically fixed N from faba beans and showed that the plant possesses a considerable environmental benefit and adaptability; it can be grown under different field and environmental conditions with the possibility of fixing between 15 [5] and 648 kg N ha$^{-1}$ [6]. Among legume crops, studies have reported that faba bean (*Vicia faba* L.) and lupine (*Lupinus albus* L.) show the highest average $N_2$ fixation; therefore, these crops increase the N benefit for subsequent crops [7], representing

the best combination to obtain the desired rotation effect as a wheat-preceding crop in dry land vertisols [4,8]. Several studies have demonstrated substantial savings in the amount of N fertilizer required to maximize the yield of crops grown after faba bean. For example, McEwen et al. [9] found that the residual N benefit to winter wheat from a previous spring-sown faba bean represented a savings of 30 kg in fertilizer N ha$^{-1}$; in addition, the faba bean rotation had a beneficial effect on the yield response when compared to a wheat–wheat sequence. Nevertheless, the variation in the amount of nitrogen content in the soil for the successive crop is expected to be high. Kopke and Nemecek [2] found that this variation can be attributed to various factors including growing conditions, genotypic variations, and soil properties. BNF can also be enhanced through the inoculation of selected Rhizobia [10], as well as sustainable soil and crop management [11]. In recent years, several studies have been conducted on this topic and have reported variable results [3,12–16]. Most of these studies agreed that environmental conditions are very important for efficient BNF processes. Extreme temperature and excess or low soil moisture content, for instance, are critical to the growth of rhizobia and nodulation and therefore to N$_2$ fixation processes [17,18]. However, the ability of these two factors to enhance the BNF process is influenced by agronomical practices such as tillage systems. Therefore, the amelioration of soil conditions through the adoption of sustainable soil management practices, such as no/zero tillage, can play a fundamental role in enhancing the BNF potential of legume crops. Conservation agriculture (e.g., no/zero tillage) is a promising soil management strategy, especially in arid and semi-arid areas, that can provide favourable conditions for symbiosis [8,19], for example, through reducing soil temperature and enhancing soil water availability by reducing bulk density; increasing organic matter content, particularly in the top soil layer; and increasing aggregate stability with a balance between macro and micropores [20]. A review study by Torabian et al. [21] highlighted that adopting conservation tillage practices, such as zero tillage, can enhance soil quality when compared with conventional tillage practices that disrupt soil structure. The authors indicated that the benefits of maintaining soil structure through no-tillage systems typically include improved biological processes and increased microbial biodiversity, in particular Rhizobia, which influenced N uptake and N fixation.

Between conservation tillage systems, no-tillage (NT) systems are an environmentally friendly soil management strategy that represent a viable alternative to conventional tillage for cereal and legume production under rain-fed Mediterranean conditions [22–24]. Considering legumes in particular, the ability of NT systems to stimulate N$_2$ fixation [16] by improving soil pH, temperature, moisture, and nutrient availability, therefore enhancing the structure and function of the microbial community [25,26], has been documented. The previous implementation of NT conservation practices at the study site produced interesting results in terms of energy savings [27] and a reduction in greenhouse gas (GHG) emissions from the soil [28], confirming their potential as a sustainable alternative to conventional tillage systems. Nowadays, the selection of no-tillage systems along with a legume rotation in rain-fed Mediterranean agriculture is increasingly being adopted, especially in areas where water and soil fertility are limited. López-Bellido et al. [29] indicated that this combination of tillage and rotation is key to improving soil fertility and increasing C reserves and the rate of C accumulation by soil; therefore, this approach represents a valuable mitigation strategy in the face of climate change in the Mediterranean region.

Considering the investigations detailed above, we evaluated the potential improvement in BNF due to faba bean crop rotation with conservative soil management techniques. The objective of the present study was to evaluate the N$_2$ fixation of faba beans grown in a rotation with durum wheat under three soil management systems. We aimed to test the ability of the no-tillage soil management technique to contribute to the enhancement of the N$_2$ fixation process by the faba bean crop under rain-fed Mediterranean conditions.

## 2. Materials and Methods

### 2.1. Site Description

The study was carried out at the Experimental Education Centre "E. Pantanelli" of the University of Bari, located in a typical Mediterranean area in Policoro (Basilicata region (MT), southern Italy; 41°7′31″ N, 16°52′0″ E). The site is located 15 m above sea level and is characterized by a Mediterranean climate according to the De Martonne classification, with an average annual rainfall of 560 mm distributed mainly during autumn and winter, a mean maximum temperature range between 12.8 °C and 31 °C, and a mean minimum temperature range between 4.0 °C and 11.3 °C.

In the study area, the soil is more than 1.2 m deep. The soil's physical characteristics (sand 398 g kg$^{-1}$, silt 374 g kg$^{-1}$, and clay 228 g kg$^{-1}$) were measured according to the pipette procedure [30]; the results indicated that the soil had a loamy texture, according to the USDA classification system. The soil water retention curve, which was determined in the laboratory using porous plate funnels and Richard's pressure chambers [31], determined that the soil moisture at field capacity was 0.32 m$^3$ m$^{-3}$ and that at the wilting point was (−1.5 MPa) 0.15 m$^3$ m$^{-3}$ of the dry soil weight. Other soil characteristics, including bulk density (1.25 kg dm$^{-3}$), were measured using the core method [32]. The measurements were obtained according to the Official Methods of Soil Analysis (USDA). For the analysis, the following parameters were obtained: total nitrogen (Kjeldahl method); available P (Olsen method); exchangeable K$_2$O (ammonium acetate method); nitrate (nitrate test kit NO$_3$$^-$); organic matter (Walkley–Black method); pH (Conyers and Davey, 1988); and active carbonate (Druineau). Table 1 shows the average values of the soil characteristics in the study area.

**Table 1.** Soil chemical properties at the experimental study site.

| Characteristics | M. Unit | Value |
|---|---|---|
| Total nitrogen | g kg$^{-1}$ | 1.7 ± 0.22 |
| Available P | mg kg$^{-1}$ | 27.6 ± 0.51 |
| Nitrate | ppm | 14.3 ± 0.39 |
| Organic matter | % | 2.8 ± 0.22 |
| Total lime | % | 8.8 ± 0.39 |
| Exchangeable Na (ESP) | % | 1.9 ± 0.15 |
| pH | - | 7.72 ± 0.17 |
| Exchangeable K$_2$O | mg kg$^{-1}$ | 227 ± 5.0 |
| Total carbonate | g kg$^{-1}$ | 15.0 ± 1.2 |
| Active carbonate | g kg$^{-1}$ | 5.0 ± 0.18 |
| Soil salinity (ECe; saturated paste extract method) | dS m$^{-1}$ | 0.95 ± 0.03 |

### 2.1.1. Crop Management

The experiment was implemented over two growing seasons (2012/2013 and 2013/2014) within a long-term durum wheat–faba bean rotation experiment (Figure 1) started in 2009. The tillage system was the main treatment, with reduced (RT), conventional (CT), and no-tillage (NT) systems; wheat was used as reference crop for N$_2$ fixation. Faba bean cv Prothabat 69 was sown at a rate of 50 germinable seeds m$^{-2}$ (about 180 kg ha$^{-1}$) in 35 cm wide rows, while durum wheat cv Iride was sown in 18 cm wide rows at a seeding rate of 400 germinable seeds m$^{-2}$ (about 200 kg ha$^{-1}$). In both years, crops were sown on the 24 November 2013 and on the 26 November 2014 and harvested 210 days after planting (DAP) for faba bean and 225 DAP for durum wheat. The sowing was carried out using different sowing machines by opening a narrow slot. In the NT system, the crop was sown in unprepared soil using a sod seeder (IGEA 2700 La Semina Sodo) to obtain the right seed coverage without soil preparation. In the CT and RT systems, sowing was performed using a conventional seed driller (IMA LAROCCA) after essential soil disturbance was complete. The three soil management systems were described in detail in a previous study [27]. Briefly, the NT system refers to a method of planting the crop in unprepared soil using a direct drill seeder while the RT system refers to the use of a subsoiler at a depth of

20 cm in late August followed by disc harrowing at a depth of 15 cm for soil bed preparation before sowing. In contrast, the CT system involves a deeper ploughing in late August using mouldboard at a depth of 35 cm followed by disc harrowing at a depth of 15 cm depth as essential seed bed preparation before sowing.

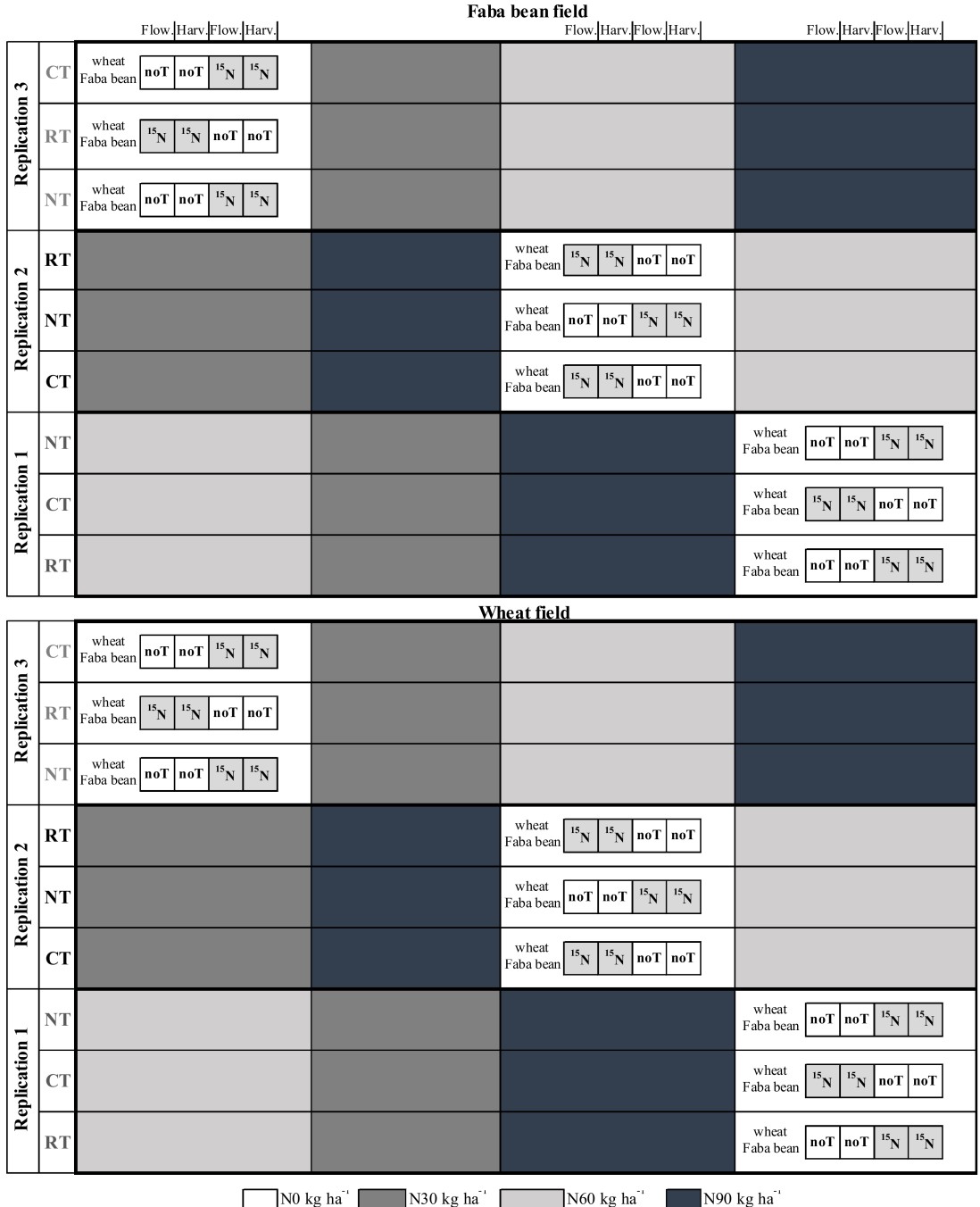

**Figure 1.** Schematic overview of $N_2$ fixation experimental design and microplot positions within the long-term wheat–faba bean rotation experiment. The microplots were organized within the main rotation experimental plot, which received 0 kg ha$^{-1}$ of nitrogen during wheat cultivation. The size of the main plot was 1 ha, while the microplots were 1 m$^2$; nitrogen at levels of 0, 30, 60, and 90 was applied to the subplot of the main rotation experiment during wheat cultivation. CT, RT, and NT stand for conventional tillage, reduced tillage, and no tillage, respectively; noT: no treated microplots; $^{15}$N: 15N-enriched microplots. Flow. and Harv. indicate the corresponding microplots that were harvested at flowering and physiological maturity, respectively.

For weed control in the no-tillage plots, glyphosate (N-(phosphonomethyl) glycine) was used at a rate of $1.5 \, l \, ha^{-1}$ before sowing in both crop fields. Concerns related to the possible effect of glyphosate on the microbial community, and, therefore, nitrogen fixation ability, of the faba bean crop were previously addressed by Weaver et al. [33], who found that glyphosate caused no meaningful shifts in the whole microbial community even when applied at greater than label rates. For tilled plots, weed control was performed using a mix of Mesosulfuron-Methyl 3%, Iodosulfuron-Methyl-Sodium 0.6%, and Mefenpyr-Diethyl 9% at doses of $400 \, g \, ha^{-1}$ in the wheat field; in the faba bean field, weeds were controlled by applying Pendimethalin before emergence and after sowing at doses of $455 \, g \, L^{-1}$, while Propaquizafop was applied during the stem elongation phase at a concentration of $100 \, g \, L^{-1}$.

Fertilizer application was carried out using $150 \, kg \, ha^{-1}$ of simple superphosphate before sowing in the faba bean field, while in the durum wheat field, with respect to the nitrogen fixation experiment, the only source of nitrogen was 15N sulphate.

2.1.2. Soil Analysis

The total organic matter (OM) content was determined using the Walkley–Black method [34]. Since the formation and stability of organic matter are largely related to long-term moisture and temperature trends, OM content was analysed at the end of each growing season in each year. The water stability index (WSI) was calculated as a frequently used indicator of soil quality. To determine the stability of soil aggregates, a wet-sieving method was used [35]. Briefly, air-dried soil aggregates (1–2 mm), collected from the surface layer (0–100 mm) of the main plots under different tillage systems, were placed on a 0.25 mm mesh sieve and moistened by a capillary rise from a layer of wet sand. They were then immersed in deionised water and shaken with an alternate vertical movement (30 times $minute^{-1}$) at room temperature. The WSI was then calculated as follows:

$$\frac{\mathbf{B} - \mathbf{C}}{((\mathbf{A} * \mathbf{K}) - \mathbf{C})} * \mathbf{100} \tag{1}$$

where A is the mass (g) of air-dried soil aggregates; B is the mass of oven-dried aggregates that remained in the sieve; C is the mass of the sand fraction; and K is the correction factor for soil moisture content (k = mass of oven-dried aggregates divided by the mass of air-dried aggregates).

*2.2. N Fixation Experiment and $^{15}$N-Labelled Fertilizer Treatments*

Two concurrent $^{15}$N enrichment experiments were carried out in both the faba bean and durum wheat fields. The main rotation experiments consisted of three tillage treatments (CT, RT, and NT) as the main treatment and four levels of nitrogen (N0, N30, N60, and N90 kg of nitrogen $ha^{-1}$) as a subplot treatment, organized according a strip plot design, repeated three times.

The $^{15}$N enrichment experiment consisted of microplots inside the main experimental plot that were established within the N0 sub-treatment of each tillage system in both the wheat and faba bean fields each year (Figure 1).

As reported in Figure 1, tillage system, species treatment, and harvest time were considered in the microplots for the nitrogen fixation experiment (3 tillage systems × 2 harvest times (flowering and grain maturity) × 2 treatments (treated and test) × 2 species (wheat and faba bean) × 3 replications). The treated microplots were fertilized using $^{15}$N-enriched ammonium sulphate (2.5 atom % excess $^{15}$N; Spectra 2000 Srl) at a rate of 3.71 g $microplot^{-1}$ equal to 37.1 Kg $ha^{-1}$ of N during the first half of February (corresponding to the formation of side shoots in fava beans and the tillering stage in wheat) each year (Figure 1). Treatments with $^{15}$N-enriched ammonium sulphate were realised via application to the soil surface of the microplot area in 4 L of distilled water per microplot using a hand sprayer.

*2.3. Data Collection and Analysis*

The yield parameters (straw and/or grain) of both crops were collected each year at the flowering and physiological maturity stages from the three replicates (Figure 1). At the flowering stage, the total aboveground biomass (stem + leaves) was collected from the entire area of the microplots for each species. Dried biomass was measured at 60 °C to obtain a constant weight (Ruisi et al., 2012). Similarly, the yield parameters (grain and straw) were collected from the entire area of the microplots (Figure 1) at maturity.

For the analysis, plant materials from each microplot (total aboveground biomass at the flowering phase and straw biomass and grain biomass at the harvesting phase) were milled (Retch mill) and passed through the grid of a 0.2 mm sieve. The total N and % $^{15}$N atomic excess of plant samples were analysed at the laboratory of Iso-Analytical Limited (Cheshire, UK). The nitrogen isotope analysis was carried out using elemental analyser isotope ratio mass spectrometry (EA-IRMS).

*2.4. Estimating $N_2$ Fixation*

The percentage of nitrogen fixed from the atmosphere (%Ndfa) in faba bean crops was estimated for the growing seasons 2012–13 and 2013–14 using different dilutions of 15N-fertilizers (López-Bellido et al., 2006).

2.4.1. Calculation of the Proportion of $N_2$ in Faba Bean (%Ndfa)

Using the $^{15}$N dilution, the share of $N_2$ fixed by faba bean (Ndfa%) in $^{15}$N-labelled soil was estimated at both the flowering and physiological maturity stages.

The $^{15}$N dilution methodology used, as described by López-Bellido et al. [22] and Van Kessel and Hartley (2000), applies the following:

$$\%\textbf{Ndfa} = \left[ 1 - \frac{\textbf{atom\%}^{15}\textbf{N excess faba bean}}{\textbf{atom\%}^{15}\textbf{N excess wheat}} \right] \times 100 \tag{2}$$

where atom% $^{15}$N excess = (%atom $^{15}$N sample (faba bean and wheat) − %atom $^{15}$N $N_2$ air) and atom% $^{15}$N of air $N_2$ = 0.3663.

The mean atom% $^{15}$N excess in $^{15}$N-labelled wheat was used as a reference in the %Ndfa calculation for each of the $^{15}$N-labelled faba bean treatments. The total amount of N fixed (Ndfa) by the faba bean crop was determined by multiplying the %Ndfa by the total N found in the shoots (mg plant$^{-1}$).

2.4.2. Calculation of the Amount of N Fixed by Faba Bean

The amount of $N_2$ fixed was determined using the obtained %Ndfa and N content in the faba beans from each replication. The average was then calculated using the methodology described by López-Bellido et al. [29], as shown in the following Equation (3):

$$\textbf{N}_2 \textbf{ fixed} \left( \textbf{kg ha}^{-1} \right) = \frac{\left[ \%\textbf{Ndfa} \times \textbf{total N in faba bean} \left( \textbf{kg ha}^{-1} \right) \right]}{100} \tag{3}$$

where total crop nitrogen is the result of the faba bean dry matter weight (kg ha$^{-1}$) multiplied by the nitrogen content (%) in the dry matter [36].

*2.5. Crop N Uptake and Soil N Balance*

Firstly, crop N uptake (kg N ha$^{-1}$) was calculated for both faba bean and wheat crops during the flowering and maturity stages by multiplying the N content (%) in plant biomass (seeds and/or straw) by the relative production amount (t ha$^{-1}$). Secondly, soil N balance [37] was calculated as the difference between fixed $N_2$ and N removal in faba bean seeds [22], as in the following Equation (4):

$$\textit{Soil N balance} = \textit{N}_2\textit{fixed} - \textit{seed N} \tag{4}$$

*2.6. Statistical Analysis*

The data were subjected to an analysis of variance (ANOVA) test. In addition, treatment means were compared using the Student Newman–Keuls test at a 5% probability level. The CoStat software was used for the analysis. Angular transformations were performed as necessary to normalise the variance before analysis. The tillage system was considered as the main factor, while the year was considered as a random effect according to Gomes and Gomes [38].

*2.7. Weather Conditions*

Weather data were obtained from an in situ weather station installed approximately 500 m away from the study field. The mean minimum and maximum temperatures and average monthly rainfall (mm) during the study years are reported in Figure 2 and compared to the 50-year average. The figure shows the minimum (Tmin) and maximum (Tmax) mean temperatures and average mean temperatures during the study years compared to the 50-year average for mean monthly temperatures. In addition, the figure shows the variation in monthly rainfall amount during the two growing seasons compared to the 50-year average. In both years, an average seasonal temperature of 13 °C was reported, ranging between a minimum of 8 °C (slightly below average during December and February) and 9.6 °C (slightly above average during December and January) during the 2013 and 2014 seasons, respectively, and a maximum of 22.5 °C. The overall Tmin and Tmax values recorded during the study period were close to the 50-year average, with some exceptions. The mean annual rainfall during the 2012/13 growing season (November–June) was close (340 mm) to the mean value for the area over the last 50 years (416 mm) during the same period. During the second year (2013/14), a considerably higher amount of rainfall (710 mm) was reported when compared to the 50-year average for rainfall during the same period.

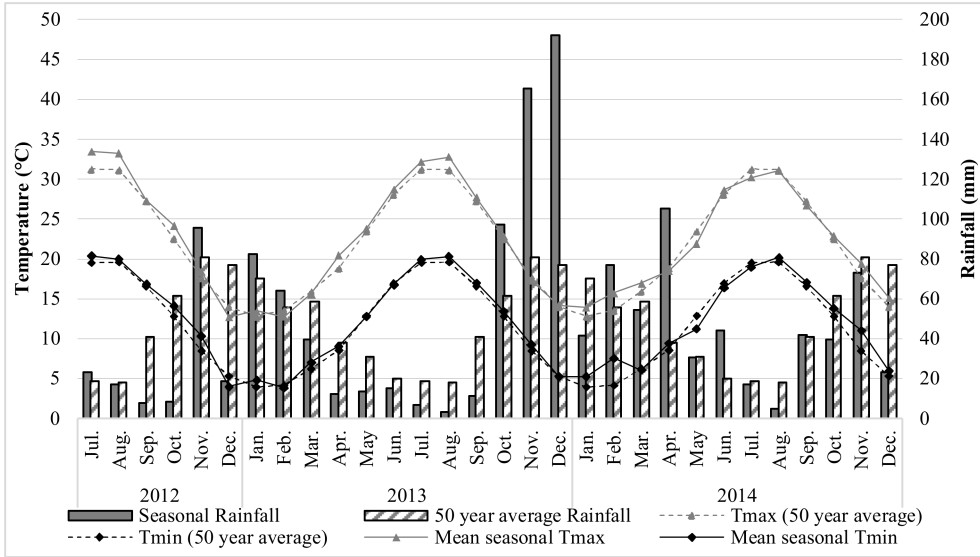

**Figure 2.** Mean minimum (Tmin) and maximum (T max) temperatures and average monthly rainfall amount during the study years (2012/2013 and 2013/2014) compared to the 50-year (1959−2010) mean minimum (Tmin) and maximum (T max) temperatures and average monthly rainfall at the E. Pantanelli educational farm in Policoro (southern Italy).

## 3. Results

*3.1. Crop Yields*

The total aboveground biomass at the flowering stage and grain and straw yield at maturity were measured for faba bean and wheat crops under the three tillage systems each year (Table 2). The data showed that the tillage system had a significant ($p \leq 0.05$) influence on the yield parameters of both crops, with some exceptions (Table 2). In general,

the tillage systems varied in their effects on biomass and grain yield production according to the growth stage. The total aboveground biomass of wheat at the flowering stage was significantly influenced by the tillage system and was higher under the NT system (Table 2); however, the tillage system had no significant effect on the total aboveground biomass of faba bean at the same stage. At maturity, the grain yield varied significantly between tillage systems. In faba bean, the grain yield was highest under CT (3.28 t ha$^{-1}$), followed by NT (2.89 t ha$^{-1}$), and RT (2.55 t ha$^{-1}$). The average wheat grain yield was significantly higher (6.48 t ha$^{-1}$) under the NT system when compared to the RT (5.37 t ha$^{-1}$) and CT (5.31 t ha$^{-1}$) systems. Total biomass (grain and straw) showed the same trend as grain yield in both crops. The overall results indicated that the second year showed significantly higher wheat biomass production (13.34 tha$^{-1}$ vs. 11.11 tha$^{-1}$), while the differences in faba bean biomass production were not significant between the two years (6.18 tha$^{-1}$ vs. 6.39 tha$^{-1}$).

**Table 2.** Faba bean and wheat aboveground biomass at flowering and straw (stem and leaves) and grain yield at maturity with respect to year and tillage system.

| Year | Treatment | Flowering Faba Bean Total Aboveground Biomass (t ha$^{-1}$) | Wheat | Maturity Faba Bean Grain (t ha$^{-1}$) | Straw (t ha$^{-1}$) | Harvest index | Wheat Grain (t ha$^{-1}$) | Straw (t ha$^{-1}$) | Total Biomass (t ha$^{-1}$) |
|---|---|---|---|---|---|---|---|---|---|
| 2013 | CT | 4.64 | 5.20 | 3.29 | 6.42 | 0.34 | 5.88 | 5.60 | 11.48 |
|  | RT | 4.65 | 5.30 | 2.36 | 6.64 | 0.26 | 5.86 | 5.15 | 11.01 |
|  | NT | 4.58 | 5.49 | 2.84 | 6.92 | 0.29 | 6.40 | 4.43 | 10.83 |
| 2014 | CT | 4.83 | 4.71 | 3.27 | 6.13 | 0.35 | 4.74 | 8.11 | 12.84 |
|  | RT | 5.02 | 5.16 | 2.75 | 5.78 | 0.32 | 4.89 | 7.79 | 12.67 |
|  | NT | 5.26 | 5.49 | 2.94 | 6.33 | 0.32 | 6.56 | 7.95 | 14.51 |
| Mean tillage (T) |  |  |  |  |  |  |  |  |  |
|  | CT | 4.74 a | 4.95 b | 3.28 a | 6.27 a | 0.34 a | 5.31 b | 6.85 a | 12.16 ab |
|  | RT | 4.84 a | 5.23 ab | 2.55 b | 6.21 a | 0.29 b | 5.37 b | 6.47 ab | 11.84 b |
|  | NT | 4.92 a | 5.49 a | 2.89 ab | 6.63 a | 0.30 b | 6.48 a | 6.19 b | 12.67 a |
|  | ns | * | * | ns | * | ** | * | * |
| Mean year (Y) |  |  |  |  |  |  |  |  |  |
|  | 2013 | 4.62 a | 5.33 a | 2.83 a | 6.66 a | 0.30 b | 6.05 a | 5.06 b | 11.11 a |
|  | 2014 | 5.04 a | 5.12 a | 2.99 a | 6.08 b | 0.33 a | 5.39 b | 7.95 a | 13.34 b |
|  | ns | ns | ns | * | * | ** | ** | * |
| Interaction |  |  |  |  |  |  |  |  |  |
| T × Y |  | ns | ns | ns | ns | ns | * | * | ns |
| **Total mean** |  | **4.83** | **5.22** | **2.91** | **6.37** | **0.31** | **5.72** | **6.50** | **12.22** |

CT—conventional; RT—reduced; N—no-tillage; Y—cropping year; T—tillage. 2013 and 2014 represent the 2012/13 and 2013/14 growing seasons, respectively. ns, *, **—nonsignificant or significant at $p \leq 0.05$, 0.01, respectively. Different letters within each column indicate significant differences according to the Student Newman—Keuls (SNK) multiple-range test ($p \geq 0.05$).

### 3.2. N Content and N Uptake

#### 3.2.1. N Content in Biomass

Across both studied years and tillage systems, the N content in plant materials at the flowering stage ranged between 3.22% and 5.23% in faba bean and between 1.89% and 3.42% in wheat (Table 3). On average, the N content in faba bean biomass was significantly higher in the CT (4.59%) and NT (4.14%) groups; in wheat, higher N content was reported in the CT (2.81%) group followed by the RT (2.72%) group, whereas the NT system resulted in only 1.97% N content. The first year showed significantly higher N content in the harvested aboveground biomass at the flowering stage for both crops. At maturity, the N content in crop biomass (grain and straw) showed an increase in both crops. In the faba bean crop, approximately 58–68% of the nitrogen was concentrated in the grain, with the highest being reported in the grain under the NT system. Similarly, greater amounts of N were accumulated in wheat grains, with the lowest value (76%) reported under CT and the highest (78%) under the NT system. However, the total N concentration (in both grain and straw) of the faba bean crop was not significantly influenced by the tillage system and the study year (Table 3). In contrast, the tillage system had a significant influence

on the N concentration in both the straw and grain of the wheat crop, with the lowest N concentration value reported under the NT system. The effect of the study year presented differences in the N concentration of straw, where the second year showed a significantly higher value (0.78% vs. 0.58%) (Table 3).

**Table 3.** Elemental N concentration (% on the dry weight) in faba bean and wheat crop aboveground biomass and grain at two crop growth stages with respect to tillage system and study year.

| Year | Treatment | Flowering | | Maturity | | | |
| | | Faba Bean | Wheat | Faba Bean | | Wheat | |
| | | Total Aboveground Biomass | | Grain | Straw | Grain | Straw |
|---|---|---|---|---|---|---|---|
| 2013 | CT | 5.23 | 3.42 | 4.84 | 1.23 | 2.60 | 0.60 |
| | RT | 4.31 | 3.32 | 4.73 | 1.39 | 2.45 | 0.79 |
| | NT | 5.06 | 2.06 | 4.94 | 1.27 | 1.73 | 0.34 |
| 2014 | CT | 3.80 | 2.20 | 4.69 | 1.81 | 2.37 | 0.90 |
| | RT | 3.23 | 2.12 | 4.80 | 1.44 | 2.38 | 0.94 |
| | NT | 3.22 | 1.89 | 4.81 | 1.48 | 2.12 | 0.50 |
| Mean tillage (T) | | | | | | | |
| | CT | 4.51 a | 2.81 a | 4.77 a | 1.52 a | 2.49 a | 0.75 b |
| | RT | 3.77 b | 2.72 a | 4.77 a | 1.41 a | 2.41 a | 0.86 a |
| | NT | 4.14 ab | 1.97 b | 4.87 a | 1.38 a | 1.93 b | 0.42 c |
| | | ** | ** | ns | ns | * | ** |
| Mean year (Y) | | | | | | | |
| | 2013 | 4.86 a | 2.94 a | 4.84 a | 1.29 a | 2.26 a | 0.58 b |
| | 2014 | 3.42 b | 2.07 b | 4.77 a | 1.58 a | 2.29 a | 0.78 a |
| | | ** | * | ns | ns | ns | ns |
| Interaction | | | | | | | |
| | T × Y | * | ns | ns | ns | ns | ns |
| Total mean | | 4.14 | 2.50 | 4.80 | 1.09 | 2.28 | 0.68 |

CT—conventional; RT—reduced; N—no tillage; Y—cropping year; T—tillage. 2013 and 2014 represent the 2012/13 and 2013/14 growing seasons, respectively. ns, *, **—nonsignificant or significant at $p \leq 0.05$, 0.01, respectively. Different letters within each column indicate significant differences according to the Student Newman–Keuls (SNK) multiple-range test ($p \geq 0.05$).

### 3.2.2. Crop N Uptake

The amount of N uptake by faba bean and wheat crop biomasses under different soil systems and at both growth stages is reported in Table 4. From the table, the overall data indicate that N uptake increased as the crops grew and was higher in crop biomass at maturity compared to flowering. At crop maturity, the N uptake by crop seeds was much higher when compared to the N uptake by straw in both crops. In both crops, the 2-year average data indicated a higher percentage (>80%) of total N uptake in wheat seeds compared to faba bean seeds (about 60%); N uptake was always higher under the CT system and lower under the NT system (Table 3).

Across tillage systems and study years, N uptake by faba bean seeds ranged between 55% and 70%, whereas this percentage was higher in wheat seeds (>80%) compared to N uptake by straw. On the one hand, the effect of tillage on N uptake was significant, where the CT system resulted in significantly higher N uptake for both crops at both growth stages, except for wheat straw at maturity (Table 4). On the other hand, the results showed significantly higher N uptake in the first year in both faba bean and wheat biomasses at the flowering stage (Table 4). N uptake by faba bean biomass during the first year was higher (224.7 kg N ha$^{-1}$) than in the second year (171.7 kg N ha$^{-1}$). Similarly, N uptake by wheat biomass was higher during the first year (155.7 vs. 105.6 kg N ha$^{-1}$) than the second year. At maturity, however, the N uptake values followed different patterns for the various biomass parameters. Higher N uptake was measured in grain, 137.7 kg ha$^{-1}$ in faba bean and 210.9 kg ha$^{-1}$ in wheat, and in CT rather than NT and RT systems.

The observed differences related not only to the grain and biomass yield measured in both years but also to the N content, which was generally higher in the CT and RT treatments of stem and grain.

**Table 4.** N uptake by faba bean and wheat crop at two growth stages with respect to tillage system and study year.

| Year | Treatment | Flowering | | Maturity | | | |
|---|---|---|---|---|---|---|---|
| | | Faba Bean | Wheat | Faba Bean | | Wheat | |
| | | Total Aboveground Biomass (kg ha$^{-1}$) | | Grain (kg ha$^{-1}$) | Straw (kg ha$^{-1}$) | Grain (kg ha$^{-1}$) | Straw (kg ha$^{-1}$) |
| 2013 | CT | 242.5 | 177.9 | 159.3 | 78.6 | 252.7 | 35.2 |
| | RT | 200.1 | 176.0 | 111.5 | 92.1 | 220.2 | 46.3 |
| | NT | 231.5 | 113.2 | 140.2 | 88.1 | 169.4 | 22.0 |
| 2014 | CT | 183.7 | 103.4 | 153.4 | 111.2 | 223.2 | 42. |
| | RT | 162.3 | 109.7 | 132.1 | 83.4 | 203.2 | 45.7 |
| | NT | 169.1 | 103.6 | 141.5 | 93.6 | 196.9 | 32.7 |
| Mean tillage (T) | | | | | | | |
| | CT | 213.1 a | 140.7 a | 156.4 a | 94.9 a | 227.9 a | 38.8 b |
| | RT | 181.2 b | 142.8 a | 121.8 b | 87.8 a | 211.7 b | 46.0 a |
| | NT | 200.3 a | 108.4 b | 140.9 ab | 90.9 a | 203.2 b | 27.3 c |
| | | ** | ** | * | ns | ** | *** |
| Mean year (Y) | | | | | | | |
| | 2013 | 224.7 a | 155.7 a | 137.0 b | 86.3 a | 214.1 a | 34.5 b |
| | 2014 | 171.7 b | 105.6 b | 142.4 a | 96.1 a | 207.8 a | 40.3 a |
| | | ** | *** | * | ns | ns | * |
| Interaction | | | | | | | |
| | T × Y | * | ** | ns | ns | ns | * |
| **Total mean** | | **198.2** | **130.6** | **139.7** | **91.2** | **210.9** | **37.4** |

CT—conventional tillage; RT—reduced tillage; NT—no tillage. ns, *, **, ***—not significant or significant at $p \leq 0.05, 0.01, 0.005$ respectively. a, b—significant differences according to the Student Newman–Keuls (SNK), multiple-range test ($p \geq 0.05$).

### 3.3. Nitrogen Fixation

Table 5 shows the results for N$_2$ fixation by faba bean (%Ndfa) at flowering and maturity, estimated via 15N dilution. Significant variation was observed in relation to both the growing year and tillage system. Over the two years and at both growth stages, the highest %Ndfa values were observed under NT cultivation, in agreement with the two-year averages. At the faba bean flowering stage, the average %Ndfa value for the three tillage methods was 64.3%, ranging from a significantly low minimum under CT (61.2%) to a maximum under NT (69.1%).

**Table 5.** Atmospheric N$_2$ fixation by faba bean (%Ndfa) at flowering and maturity with respect to tillage system and growth year.

| Year | Treatments | Flowering | Maturity |
|---|---|---|---|
| | | Ndfa (%) | |
| 2013 | CT | 64.6 | 79.3 |
| | RT | 63.5 | 73.7 |
| | NT | 72.9 | 79.3 |
| 2014 | CT | 56.8 | 80.8 |
| | RT | 61.8 | 79.6 |
| | NT | 66.4 | 79.3 |
| Mean tillage (T) | | | |
| | CT | 60.7 b | 80.0 a |
| | RT | 62.7 b | 76.7 b |
| | NT | 69.6 a | 79.3 a |
| | | ** | * |
| Mean year (Y) | | | |
| | 2013 | 67.0 a | 77.4 a |
| | 2014 | 61.7 b | 79.9 a |
| | | ** | ns |
| Interaction | | | |
| | T × Y | * | ns |
| **Total mean** | | **64.3** | **78.7** |

CT—conventional tillage; RT—reduced tillage; NT—no tillage. ns, *, **—not significant or significant at $p \leq 0.05$, 0.01, respectively. a, b—significant differences according to the Student Newman–Keuls (SNK), multiple-range test ($p \geq 0.05$).

In addition, at plant maturity, the %Ndfa values were higher in the CT and NT systems, at 80.0 and 79.3%, respectively; these values were similar to each other but different with respect to RT, at 76.7%.

Comparing year by year, the %Ndfa values were not significantly different at the flowering stage and were very close at harvesting (77.4% vs. 79.9%).

The results reported in Table 6 indicate differences in the amount of $N_2$ fixed (kg ha$^{-1}$ N) by the faba bean crop during its growth. We reported an average amount of 128.9 at flowering, while this amount increased to 181.2 kg N ha$^{-1}$ at maturity (Table 6).

**Table 6.** Variation in the amount of $N_2$ fixed (kg N ha$^{-1}$) at both faba bean growth stages with respect to tillage system and study year at the Pantanelli farm (southern Italy).

| Year | Treatments | Flowering | Maturity | Mean |
|---|---|---|---|---|
| 2013 | CT | 147.5 | 186.7 | 167.1 |
| | RT | 119.7 | 146.9 | 133.3 |
| | NT | 159.5 | 178.0 | 168.8 |
| 2014 | CT | 115.9 | 213.7 | 164.8 |
| | RT | 111.4 | 175.1 | 143.3 |
| | NT | 119.2 | 186.4 | 152.8 |
| Mean tillage (T) | | | | |
| | CT | 131.7 a | 200.2 a | 166.0 a |
| | RT | 115.6 b | 161.0 b | 138.3 b |
| | NT | 139.4 a | 182.2 ab | 160.8 a |
| | | ** | ** | * |
| Mean year (Y) | | | | |
| | 2013 | 142.2 a | 170.6 a | 156.4 a |
| | 2014 | 115.5 b | 191.8 a | 153.60 a |
| | | *** | ns | ** |
| Interaction | | | | |
| | T × Y | * | ns | ns |
| **Total mean** | | **128.2** | **183.6** | **155.0** |

CT—conventional tillage; RT—reduced tillage; NT—no tillage. ns, *, **, ***—not significant or significant at $p \leq$ 0.05, 0.01, 0.005 respectively. a, b—significant differences according to the Student Newman–Keuls (SNK), multiple-range test ($p \geq$ 0.05).

The tillage system had a significant effect on the amount of $N_2$ fixed (kg ha$^{-1}$ N) at both faba bean growth stages. In each year, the amount of $N_2$ fixed by faba bean was significantly higher in both phases under the CT and NT systems (166.0 and 160.8 kg ha$^{-1}$ N, respectively), while the RT system showed the lowest amount of $N_2$ fixed (138.3 kg ha$^{-1}$ N). Considerable differences were measured at harvest time: CT registered 200.2 kgha$^{-1}$ N and NT 182.2 kgha$^{-1}$, compared to 161.0 kg ha$^{-1}$ for RT.

The growing year also had a significant effect on the amount of $N_2$ fixed by the faba bean crop. At the flowering stage, a significantly higher value was reported in the first year (142.2 Kg N ha$^{-1}$ vs. 115.5 Kg N ha$^{-1}$ during the second year), while a significantly higher value for the maturity stage was reported during the second year (191.8 Kg N ha$^{-1}$ vs. 170.6 Kg N ha$^{-1}$). This behaviour is likely related to the different meteorological conditions between flowering and harvesting, where they were more favourable in 2013/2014 season (fresh period, more rain) than during the 2012/13 season (dry period).

Nitrogen (N) Balance

N balance depends on the $N_2$ fixed by faba bean and the N removed from faba bean seeds, considering that the straw is incorporated into the soil. In this study, an analysis of our data showed that the N balance was always positive and varied significantly between tillage systems and study years (Figure 3), with an average value of 41.6 kg N ha$^{-1}$. Across the study years and tillage systems, the level of residual N in the soil was comparable between the CT and NT systems, at 43.9 and 41.3 kgha$^{-1}$, respectively, while the minimum value was measured in the RT system (39.3 kgha$^{-1}$). Considering the effect of growing

year, the 2013/14 data showed a significantly higher N balance when compared to the data from the first year (49.4 vs. 33.6 kg N ha$^{-1}$), mainly due to the significantly higher amount of N$_2$ fixed during the second year.

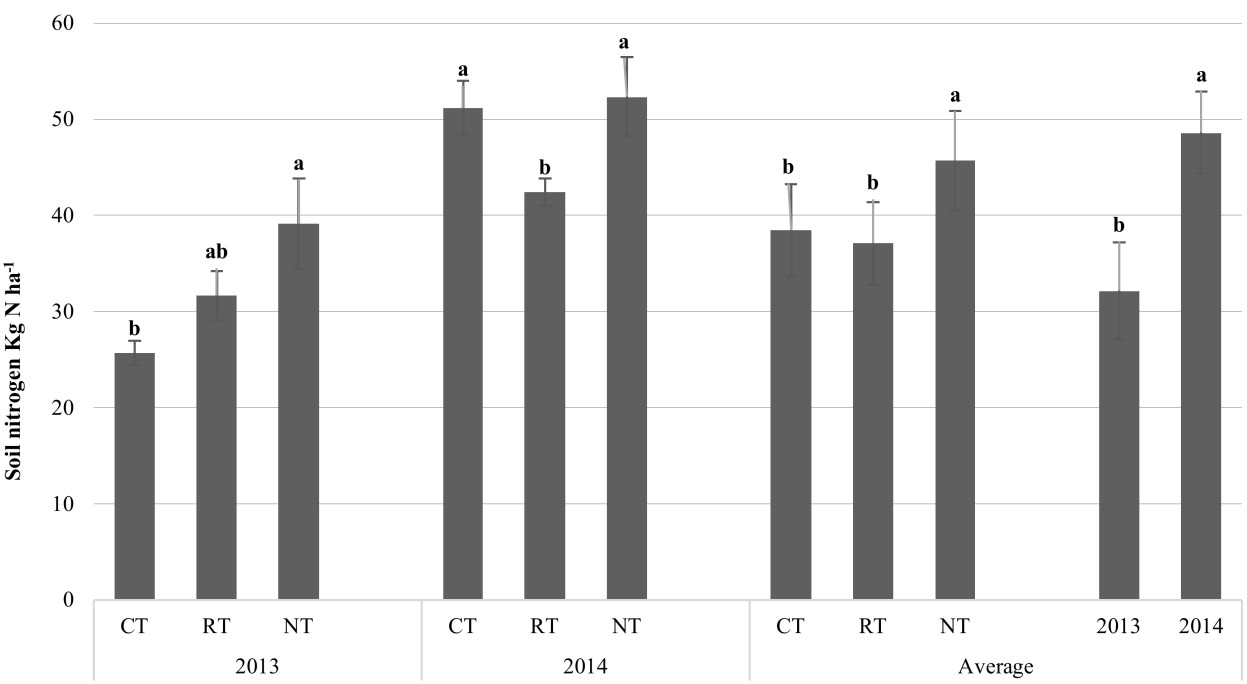

**Figure 3.** Soil nitrogen balance with respect to tillage system (CT—conventional; RT—reduced; NT—no tillage) and study year; 2013 and 2014 represent the 2012/13 and 2013/14 growing seasons, respectively. Different letters within each group indicate significant differences according to the Student Newman–Keuls (SNK) multiple-range test ($p \geq 0.05$). Each treatment was repeated three times.

### 3.4. Soil Quality Parameters

The quality of the soil was evaluated in terms of its organic matter content and stability index. An analysis of the data showed that the two measured soil quality parameters were significantly affected by the tillage system employed (Figure 4). From the figure, the reported data showed that the average (across years) organic matter content was highest in the NT soil (3.26%) followed by the RT soil (3.02%), whereas the organic matter content was significantly lower in the CT soil, at 2.89%. No significant differences were found between the two growing years in terms of organic matter content. The stability index values were found to be influenced by the year, with higher values during the first season (2012/13) (9.14%) than in the 2014/15 season (7.68%). The effect of tillage followed the same trend as organic matter content, with significantly higher values reported in the NT soil (9.18%) with respect to the RT (8.18%) and CT (7.88%) soils. The stability index is an important agronomical parameter for the physiological processes that take place in this kind of soil, considering that its high silt content and good structural conditions create an optimal habitat for soil organisms, which can increase microbial activity in soil.

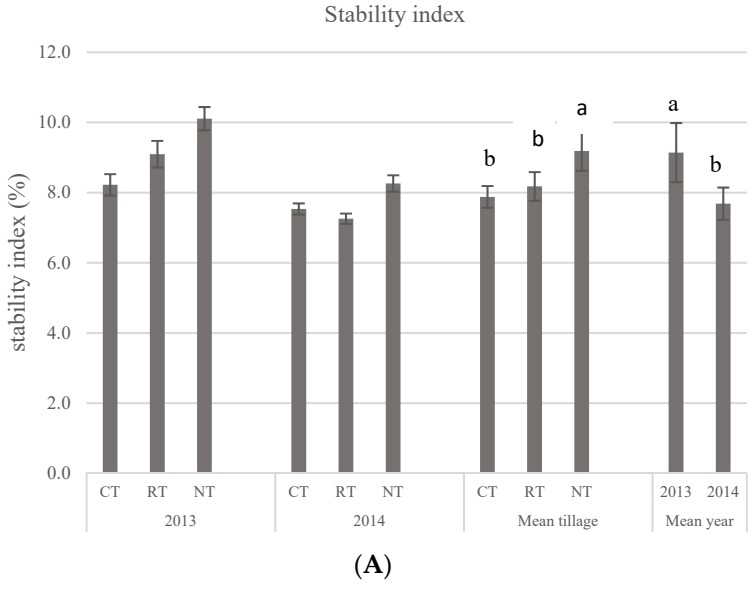

**(A)**

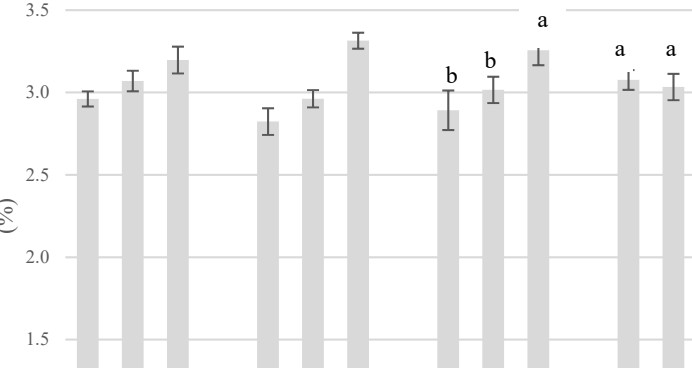

**(B)**

**Figure 4.** (**A**) Stability index and (**B**) organic matter content in the soil with respect to tillage system (CT—conventional; RT—reduced; N—no tillage) and study year at E. Pantanelli experimental farm in Policoro (southern Italy). Different letters within each group indicate significant differences according to the Student Newman–Keuls (SNK) multiple-range test ($p \geq 0.05$).

## 4. Discussions

### 4.1. Yield Results

The reported results confirmed the positive influence of no-tillage systems after the transition phase in the soil, showing a productive response in terms of total biomass and grain production.

The 2-year total biomass production in faba bean was comparable between the NT and CT (9.52 vs. 9.88 t ha$^{-1}$) systems, as was grain production; our values were in the range of those from other studies performed in a Mediterranean environment [39]. This is a comfortable indication of the effectiveness of the NT system, which produced similar results to the conservative system for faba bean cultivation with the addition of high energy savings [40]. The average faba bean yield (2.91 t ha$^{-1}$) reported in this study was higher than that reported in other studies under Mediterranean conditions, ranging between 1.84 t ha$^{-1}$ [41] and 2.02 t ha$^{-1}$ [42]. In terms of a comparison between soil management, similar results were reported by Mazzoncini et al. [43], who found that the average grain

yield of faba bean was significantly lower under NT conditions when compared to CT; however, the differences between them were small and not significant with respect to the 12-year average.

*4.2. Crop N Uptake and N Balance*

The N uptake data were in line with biomass and grain production, the main components responsible for nitrogen removal. N uptake was slightly influenced by seasonal trends and was higher during the second year (Table 4). These data are confirmed by previously reported data on grain N content for wheat [24] and faba bean [44], where the use of a CT system resulted in significantly higher N content compared to the NT system. Interestingly, low nitrogen content was found in the tissues of faba bean and straw with NT management, an aspect that was also found in the protein content of wheat in the same long-term experiment and is likely linked to the low soil temperature typical of NT systems.

Concerning N balance, the average N balance value reported in the present study was in agreement and within the range of values reported by Evans et al. [37] for N balance in faba bean crops. The authors indicated a wide variation in N balance in the faba bean crop, perhaps due to environmental conditions and the amount of $N_2$ fixed as a result of genetic variation in potential $N_2$ fixation capacity [45]. This may also include other factors, such as cropping systems and management practices. In another legume crop, Aslam et al. [46] found an N balance of 28 kg N ha$^{-1}$ for chickpeas in a rain-fed chickpea–wheat rotation. However, López-Bellido et al. [29] found that the amount of N fixed in faba beans was much greater than that in chickpeas. Taken together, the various estimations of N balance suggest that faba bean may have a more positive role in sustaining soil N fertility than other legumes due to the higher amount of $N_2$ fixation. Taking into account that the present study did not consider the contribution of roots, the positive effect of faba bean on the N enrichment of soil was clear and influenced by soil N status in rain-fed Mediterranean vertisol. The differences in N fixation appear to be linked to the moisture content in the soil, as reported by several studies [16,22,47]. These data are in line with other studies (e.g., [43]) that estimated the amount of residual N in the soil for the following cereal cultivation as up to 50–75 kg ha$^{-1}$.

*4.3. Nitrogen Fixation by Faba Bean*

The amount of $N_2$ fixed is determined by the relative reliance of a crop upon $N_2$ fixation for growth (i.e., the proportion of crop N derived from atmospheric $N_2$, %Ndfa) and the amount of N accumulated by the crop over the growing season [1]. In this study, both $N_2$ fixation parameters were estimated using 15N-labelled fertilizer during the two faba bean growth stages to evaluate the changes that occurred in the fixation process as the crop grew. The overall results indicated that both the %Ndfa and the amount of $N_2$ fixed (kg N ha$^{-1}$) were significantly increased at crop maturity. In chickpea, Turpin et al. [47] reported contrasting results for %Ndfa, finding that %Ndfa declined from 91% to only 65% at maturity due to the cessation of $N_2$ fixation during grain fill because of drought or early nodule senescence. Averaged across the years, the effect of tillage was observed at both growth stages and was always in favour of the NT system with respect to the %Ndfa and the amount of $N_2$ fixed (Tables 5 and 6). The average value of $N_2$ fixed was higher under the NT system, presumably due to favoured soil conditions and/or a higher number and dry weight of nodules plant$^{-1}$ than in the CT system [29]. Several studies (e.g., [5,12,16,48–51]) have reported faba bean $N_2$ fixation in dryland agriculture to range between 50 and 200 kg N ha$^{-1}$, with a mean value between 90 and 120 kg N ha$^{-1}$ in terms of only the aboveground biomass. In this study, $N_2$ fixation values at both the flowering (121.9 Kg N ha$^{-1}$) and maturity (178.2 Kg N ha$^{-1}$) stages were within the reported range. Similar results for $N_2$ fixation were reported by López-Bellido et al. [29], who found the amount of $N_2$ fixed to be significantly higher under the NT system when compared to the CT system. The authors justified this significant difference by the fact that the NT system offers more favourable conditions for the

stimulation of $N_2$ fixation, creating optimal conditions in the soil for microbial activities—especially rhizobia colonisation—as it preserves more humidity and favourably alters the soil temperature. In a previous study at our experimental site, De Mastro et al. [52] investigated the effects of different tillage systems on microbial soil communities and found no significant difference in the effect of different tillage systems on microbial abundance; however, the abundance of soil microbial organisms was positively correlated with the content of soil organic carbon, total nitrogen, and available phosphorus.

Our results are in agreement with the findings in the literature for the stimulation of $N_2$ fixation under the NT system; the reason for this stimulation might be attributed to the fact that NT systems can increase available N in the soil and, in turn, stimulate N demand and $N_2$ fixation. The results from a previous study [24] indicated that the lower amount of $NO_3$-N under the NT system, when compared to CT and RT, improved N availability to the crop through the enhancement of the $N_2$ fixation ability of faba bean. Previous findings from Hughes and Herridge [53] confirm these results, where the authors found that reduced amounts of soil $NO_3$-N in NT soil increased the $N_2$ fixation ability of legumes.

## 5. Conclusions

This study highlighted the role of the faba bean as natural source of nitrogen in arid/semi-arid soils, which is of particular interest to sustainable agriculture. This work gave additional value to the type of cultivation, since the positive combination of a conservative system, such as a no-tillage system, is known for its ability to reduce input and enhance/improve the BNF of faba bean crops in the rain-fed Mediterranean cereal production area.

Our results indicated that the contribution of nitrogen from faba bean to the soil was generally positive and increased significantly, with an accumulation value of up to $48$ kg ha$^{-1}$ of nitrogen in the second year. We found that the tillage system (NT vs. CT) significantly affected the %Ndfa and the amount of $N_2$ fixed by faba bean; both values were higher in the NT system.

The higher amount of $N_2$ fixation by faba bean under the NT system resulted from an improvement in several soil properties, such as increased water content and higher organic matter content. This last component can be also positively influenced by the soil condition, and the stability index represents a measure of soil quality. In addition, the activities of the soil microbial community also contributed to this improvement.

Therefore, the NT soil management technique represents a good alternative in the management of soil fertility, especially if combined with a legume crop in rotation. The results from this study may open the door for further long-term investigations of the role of local agriculture practices in enhancing BNF for sustainable crop production and the accurate estimation of the additional role of root biomass and its contribution to total soil nitrogen.

**Author Contributions:** L.T.: methodology, data curation, software, and writing—review and editing; S.A.A.: investigation and writing—original draft preparation; G.D.M.: supervision. All authors have read and agreed to the published version of the manuscript.

**Funding:** This research received no external funding.

**Conflicts of Interest:** The authors declare no conflict of interest.

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
