# Peer review of "The Effect of Tillage on Faba Bean (Vicia faba L.) Nitrogen Fixation in Durum Wheat ((Triticum turgidum L. subsp. Durum (Desf))-Based Rotation under a Mediterranean Climate"

_agronomy, doi:10.3390/agronomy13010105_

Round 1
Reviewer 1 Report
The manuscript sounds good and well-written except in places where repetitions were noticed. Methodology, especially the design of the experiment (main experiment and subplot experiments), needs more clarification. The author can consider slightly modifying the results section. Very often, the authors interpreted significantly higher or lower. It is encouraged the authors can avoid interpreting non-significant data in the results and discussion sections. The discussion section looks more like a bit of repetition of results. This section should relate the results with the available findings!
The authors can consider the reviewer comments indicated in the pdf manuscript for further improvement of the paper.

Author Response
Thank you for this second opportunity to improve our manuscript. We have carefully read all the remarks and suggestion, and the paper has been modified according your suggestion. We are sure the now the paper is better improved according the suggestion.
Attached we send list of revision applied to manuscript. English has been reviewed by an expert review system. We are sure that the paper meet the criteria of this important journal
Best regards

Reviewer 2 Report
The article analyzes a practically important issue. It is important to understand that the determined patterns greatly depend on the microclimatic and soil characteristics of the research site. The correctness of the research methodology is important, and the detailing in the “Material and Methods” section shows the correctness of the study in general.
However, there are significant shortcomings that need to be corrected.
1. It is better to mention the Latin name of the species in the title of the article.
2. It would be better not to introduce the system of abbreviations in the abstract (separately from the article, the abstract will be placed in many databases; the abstract should not not scare away readers with numerous unnecessary abbreviations).
3. Superscript and subscript characters in the article are formatted incorrectly in most cases.
4. I do not recommend placing a reference to the literature in Table 1. It would be better to provide it in the text. The data for table 1 were probably obtained as a result of several measurements. They need to be presented in the form of mean + - standard deviation.
5. Inside the formulas (line 220, 231, 240), it would be best to indicate the characteristics in the form of symbols A, B, C, D, E, and describe their meaning and units of measurement under the formulas.
6. The method used by the authors to compare samples implies their normal distribution (“The assumptions of the Newman–Keuls test are essentially the same as for an independent groups t-test: normality, homogeneity of variance, and independent observations. The test is quite robust to violations of normality" https://en.wikipedia.org). However, the authors do not write anything about checking the normality of each of the samples. The phrase "Angular Transformations were performed, as necessary, to normalize the variance before analysis" is not enough. Therefore, the readers would not be completely sure of the reliability of the patterns determined by the authors.
7. In my opinion, lines 249-270 are the Material and Methods, and not the result highlighted in the title of the article. This fragment would better suit the Material and Methods.
8. In the name of the region of the country, it is better to write both words with a capital letter, for example, line 292 - Southern Italy.
9. The values in tables 2, 3, 4 do not allow to evaluate the variability of characteristics. It is necessary to present them in the form of mean + - standard deviation. Why are only individual lines compared to each other? Why are there no performance comparisons for 2013 and 2014?
10. Empty cells in tables are not allowed, for example, in tables 2, 3, 4.
11. Tenths are separated by a dot, not a comma; incorrect units of measurement are specified (for example, lines 278, 288).
12. There is no need to write a title above the figure (line 439, 464). No frame is needed around the figure. Name of the y-axis should be added. In the title of figure 3, repetition must be indicated (despite the fact that it is indicated in different subsections of the Material and Methods).
13. Figure 4 should have a common title followed by a colon and subheading for Figures A and B. Statistical analysis in Figure 4 is unclear. In the absence of a mean + - standard deviation, how can one determine the differences between the samples?
14. In the list of literature, the names of journals are given both in full and in shortened forms, with capital letters, as well as small. There are also numerous errors in punctuation in many literature sources.
After correcting numerous technical shortcomings, I believe that the article could be recommended for publication.
Author Response

(The authors gave the same response as above.)

Reviewer 3 Report
Reviewers' comments:
Faba bean is a legume grown on a large area in the world, and how to exploit the biological nitrogen fixation role of legumes in the total ecosystem has received much attention in the Mediterranean region, but the exact amount of nitrogen fixation and the mechanisms of influence are still uncertain. Especially on the scale of crop rotation with grasses. On the other hand, conservation agriculture, such as no-till/no-tillage, is a promising soil management strategy, especially in arid and semi-arid areas, which can provide favorable conditions for plant growth, especially in top soils, This is a very interesting and timely study that integrates the evaluation of sustainability performance indicators of faba bean-wheat rotations, considering N balance, and N sources, by optimizing tillage practices and rotation. The study is very interesting and timely. It can provide relevant recommendations for policy makers and stakeholders to promote faba bean production. Overall, the manuscript is well organized and readable, but the research methods and methodology of the experiment are difficult to interpret. The manuscript is better suited to the scope of the journal. As of now, there are still several issues that need to be addressed and therefore I recommend major revisions.
1. First, several different nitrogen fertilizer levels were set in this paper, but they were less described in the results of this study. The biological nitrogen fixation effect of broad bean was only measured under the condition of no nitrogen fertilizer application. In the actual production, whether a certain amount of fertilization can guarantee the high yield or not, on the other hand, a small amount of nitrogen fertilizer application can also promote the biological nitrogen fixation of legumes such as broad bean.
2. Different isotope dilution methods involving different reference crops may produce different measurements, and it is suggested to compare and discuss the differences arising from the determination of biological nitrogen fixation in winter wheat as a reference crop. In principle, we recommend the use of non-nodulating lines of the same species for the determination of biological nitrogen fixation as a way to guarantee approximate growth conditions and nitrogen uptake patterns.
3. Nitrogen balance recommendations coupled with locally appropriate N deposition values and seed input N. If available, soil nitrate N can be measured at 0-100 cm depth below ground for analysis, which may be more convincing for interpretation of nitrogen use efficiency in legume biological nitrogen fixation or cropping systems.
Author Response

(The authors gave the same response as above.)

Reviewer 4 Report
The manuscript describes the variation of nitrogen fixation ability of broad bean under different tillage patterns. This is an important topic, many related studies have been published. However, this article neither made enough innovative scientific discoveries, nor provided novel insights into previous studies.
There are still many small problems in the manuscript, such as
Line 29 : N2 should be replaced by N2
Line 30 : in the plant tissues,. Punctuation marks with ', '
Line 39 : 4 ) For this change to ( 4 )
Line 46 and Line 59 : Inconsistent format of references in introduction and discussion, ( 8 ; 4 ) and ( 17 ; 18 ), which of the middle symbols is correct
All reports hope to be modified like Table1, remove the surrounding box.
Table6 needs to be modified, there are overlaps
Abstract : ' Data on organic matter content and stability index were higher under the NT system providing favourable conditions for microbial activities, therefore. improving the N2 fixation by faba bean. ' No analysis of microorganisms, such a conclusion is not appropriate.
Conclusion too long, need to complete the rewrite, not condensed the results of this article
Author Response

(The authors gave the same response as above.)

Round 2
Reviewer 3 Report
The manuscript has improved a lot compared to the previous version. I have no further comments.
Reviewer 4 Report
After the author modified, I think we can consider publishing